# The Management of Ocular Surface Squamous Neoplasia (OSSN)

**DOI:** 10.3390/ijms24010713

**Published:** 2022-12-31

**Authors:** Clarice H. Y. Yeoh, Jerome J. R. Lee, Blanche X. H. Lim, Gangadhara Sundar, Jodhbir S. Mehta, Anita S. Y. Chan, Dawn K. A. Lim, Stephanie L. Watson, Santosh G. Honavar, Ray Manotosh, Chris H. L. Lim

**Affiliations:** 1Yong Loo Lin School of Medicine, National University of Singapore, Singapore 119228, Singapore; 2Lee Kong Chian School of Medicine, Nanyang Technological University, Singapore 639798, Singapore; 3Department of Ophthalmology, National University Health System, Singapore 119228, Singapore; 4Duke-NUS Graduate Medical School, Singapore 169857, Singapore; 5Singapore Eye Research Institute, Singapore 169856, Singapore; 6Singapore National Eye Centre, Singapore 168751, Singapore; 7Histopathology, Pathology Department, Singapore General Hospital, Singapore 169608, Singapore; 8Save Sight Institute, Discipline of Ophthalmology, Sydney Medical School, The University of Sydney, Sydney, NSW 2000, Australia; 9Center for Sight, Hyderabad 500034, India; 10School of Optometry and Vision Science, University of New South Wales, Sydney, NSW 2052, Australia

**Keywords:** ocular surface squamous neoplasia, management, topical chemotherapy, 5-fluorouracil (5-FU), mitomycin C (MMC), interferon alfa (INFα), surgical excision, adjuncts

## Abstract

The rise of primary topical monotherapy with chemotherapeutic drugs and immunomodulatory agents represents an increasing recognition of the medical management of ocular surface squamous neoplasia (OSSN), which may replace surgery as the standard of care in the future. Currently, there is no consensus regarding the best way to manage OSSN with no existing guidelines to date. This paper seeks to evaluate evidence surrounding available treatment modalities and proposes an approach to management. The approach will guide ophthalmologists in selecting the most appropriate treatment regime based on patient and disease factors to minimize treatment related morbidity and improve OSSN control. Further work can be done to validate this algorithm and to develop formal guidelines to direct the management of OSSN.

## 1. Introduction

Ocular surface squamous neoplasia (OSSN) encompasses a broad spectrum of neoplastic changes involving the squamous epithelium of the conjunctiva, cornea and limbus ranging from mild dysplasia, intraepithelial neoplasia (carcinoma in situ) to squamous cell carcinoma (SCC) [1].

Exposure to ultraviolet (UV) B radiation is the primary risk factor for OSSN [2]. Non-modifiable risk factors include age and male gender [3,4,5]. Modifiable risk factors include smoking, chronic trauma or inflammation, exposure to chemicals, vitamin A deficiency and local immunosuppression [1,2,4,6]. Human papilloma virus (HPV) and human immunodeficiency virus (HIV) are strongly associated with OSSN [7,8,9,10,11]. Human papilloma virus serotypes 16 and 18 are thought to be cofactors in the development of OSSN [12,13], while OSSN may be the first presentation of HIV [14,15,16,17,18]. Screening for HIV should be performed in atypical cases such as younger patients with OSSN, especially those with risk factors such as having multiple sexual partners and/or a history of sexually transmitted diseases. Interestingly, a separate study [19] found that HPV does not appear to play a significant role in the etiology of OSSN in India. Instead, it is suggested that other factors listed above, such as ultraviolet radiation and immunodeficiency, played a more important role.

Although rare, OSSN is the most common non-pigmented tumour of the ocular surface [20], with a worldwide age-standardised rate of 0.26 cases per 100,000 per year. There is a higher incidence in African countries of 3–3.4 per 100,000/year [2]. Ocular surface squamous neoplasia is becoming more common in Africa, which can be partly attributed to increased survival of HIV-infected patients [2]. A study conducted in Canada also found an increasing incidence of malignant OSSNs, likely in part due to an aging population [21].

OSSN occurs with equal frequency in both men and women in Africa [2] and in parts of Asia including Saudi Arabia [22]. However, in most of the rest of the world, OSSN is more common in men [2]. This is partly due to differing risk factors, with a higher prevalence of HIV and HPV in Africa contributing to the increased risk of developing OSSN in women [2]. Malignant OSSNs generally follow the same epidemiological pattern, with studies finding that they are more common in males in Canada, Iran, and the United States [21,23,24], and equally common among both genders in Africa [2].

Although no causative genetic mutations have been identified [25], several mutations including the tumour suppressor gene p53 [26], the telomerase reverse transcriptase (TERT) gene promoter [27], a disintegrin and metallopeptidase domain 3 (ADAM3) [28,29], matrix metalloproteinase 9 (MMP-9), matrix metallopeptidase 11 (MMP-11) and clusterin [30] have been associated with the pathogenesis of OSSN. Further, DNA hypomethylation at the DNA methyltransferase 3-like (DNMT3L) promoter has been identified in OSSN, however its physiologic significance remains unclear [31].

Clinically, OSSN typically presents as a unilateral vascularized mass, with bilateral or multifocal presentations being less common [1]. Lesion morphology ranges from gelatinous, leukoplakic, papillary, nodular to nodulo-ulcerative [32]. Nodular and papillomatous lesions are associated with higher histopathologic grade [33]. Nodulo-ulcerative lesions are rare, aggressive variants which have been described in a case series of six patients with four having intraocular extension suggesting that they may be more invasive compared to other morphologies [32].

Diagnosis of OSSN is more challenging when associated with other ocular surface lesions sharing risk factors of UV exposure such as pterygia and pinguecula [34]. Histopathological evaluation following an incisional or excisional biopsy is the gold standard for the diagnosis of OSSN [33]. Less invasive modalities include impression [35] or exfoliative cytology [36,37], in vivo confocal microscopy (IVCM) [38,39] and high-resolution or ultra-high-resolution anterior segment optical coherence tomography (HR-OCT) [40].

Surgical excision is the gold standard for the management of OSSN. Excision via Shields’ no touch technique with 4 mm margins, followed by intraoperative cryotherapy with the double freeze-and-slow-thaw technique achieved a low rate of recurrence [41], Surgical management can be associated with the development of complications such as limbal stem cell deficiency, symblepharon formation, conjunctival hyperaemia and conjunctival scarring [6,42,43,44]. Limbal stem cell deficiency may arise as OSSN typically involve the limbal region. In cases with orbital extension resulting from late presentation, delayed or missed diagnosis, and/or incomplete excision, orbital exenteration may be required [45].

Recently, there has been a shift towards medical management including topical chemotherapy drugs and immunomodulatory agents such as 5-fluorouracil (5-FU), mitomycin C (MMC), and interferon alfa-2b (IFNα−2b). Such agents have been used in combination with surgical excision but also as monotherapy due to their ability to treat the entire ocular surface [46].

A combination of surgical and medical methods has also shown to be effective in cases with high recurrence risk. Topical chemo-reduction with MMC may allow for less extensive surgical resection and tissue reconstruction [47]. Post-operative topical IFNα−2b therapy lowered the recurrence rate in patients with positive margins to a level similar to that of negative margins [48]. Topical chemotherapeutics such as IFNα−2b and MMC have been used preoperatively for tumour reduction, especially for extensive tumours which may be less amenable to monotherapy with such topical agents [47,49].

Topical IFN alfa-2a (IFNα−2a) has been used both as primary therapy and tumour reduction prior to surgical management [50,51]. Subconjunctival IFNα−2b has also been used as adjuvant therapy to reduce the risk of recurrence [52]. However, it is less commonly used compared to IFNα−2b. The two drugs differ by an amino acid present at position 23 of the protein. Lysine is present in IFNα−2a in this position, while IFNα−2b has arginine [53].

Adjunctive treatments reported to be in use include radiotherapy, topical anti–vascular endothelial growth factor (anti-VEGF) agents [54,55], topical cidofovir [56,57], topical retinoic acid [54] and photodynamic therapy [58].

Currently, there remains no consensus regarding the best way to manage OSSN. No guidelines exist to advise an approach to management. The purpose of this paper is to summarize treatment options for the management of OSSN and provide recommendations to guide ophthalmologists in selecting the most appropriate treatment regime based on patient and disease factors to minimize treatment related morbidity and improve OSSN control.

## 2. Surgical Management

Surgical excision is the gold standard for the management of OSSN. The primary method of excision has been described by the Shields group [41]. This technique recommends macroscopic tumour-free margins of at least 4 mm during surgery to increase the likelihood of complete tumour resection [41]. This is followed by intraoperative cryotherapy applied to conjunctival and limbal margins via the double freeze-and- slow- thaw technique to rupture tumour cell membranes and occlude associated feeding blood vessels [59,60]; this further reduces the risk of recurrence [48]. Keratoepitheliectomy can be performed in cases of corneal involvement. Absolute alcohol is applied for 1 min before excision with tumour-free margins of at least 2 mm [41].

For scleral invasion, a partial lamellar sclerectomy can be performed [61]. In rarer cases of intraocular invasion, enucleation [62] should be considered while orbital invasion requires exenteration [63].

Primary wound closure may be performed for small wounds while amniotic membrane coverage is preferred for larger wounds to aid healing and to minimise inflammation and scarring. Despite amniotic membrane closure, extensive excision of the conjunctiva may lead to symblepharon and conjunctival scarring [64,65]. Surgical management can also be associated with other complications such as limbal stem cell deficiency (LSCD) [6,42,43,44]. Surgically induced scleral necrosis (SINS) is a relatively rare but well-documented post-operative complication that is more likely to occur following adjuvant therapy. This can occasionally lead to devastating complications such as scleral melt and perforation [66]. A modified Mohs micrographic excision technique with intraoperative cryotherapy has been proposed to allow for maximal conservation of healthy tissue [61] while concomitant limbal epithelial transplantation has been useful in preventing LSCD [67,68,69]. However, these techniques are time-intensive and require specific surgical expertise.

Surgical excision alone may lead to tumour recurrence due to the presence of residual microscopic disease. Erie et al. [70] found that recurrence of conjunctival intraepithelial neoplasia and squamous cell carcinoma did not correlate with cell type or degree of atypia, but with the presence or absence of positive margins. Recurrence rates with positive margins can be as high as 56% and were reduced to 33% with negative margins [71]. Recently, lower recurrence rates of 0–21% have been reported, likely due to the use of intraoperative cryotherapy, adjunctive postoperative topical MMC and postoperative topical INFα−2b in patients with positive margins [48,72,73]. Thus, to reduce recurrence rates, adjunctive topical chemotherapy with IFNα−2b, 5-FU or MMC should be performed if histopathological evaluation shows positive margins.

Although surgical management potentially shortens the overall treatment period compared to medical treatment, the complications highlighted above may limit its wider application. As proposed by Karp [74], surgery may be preferred for small (<4 clock hours (Shields) [41] or ≤5 mm (American Joint Committee on Cancer (AJCC)) [75]), unifocal lesions, lesions with an uncertain diagnosis and first presentations of OSSN. Considerations of factors such as accessibility and affordability of health services and medications, tolerance of and compliance with treatment, patient comorbidities and preferences also influence the choice between surgical and medical management.

## 3. Medical Management

To avoid the complications of surgery, there has been a shift towards medical management [76] which uses both topical chemotherapy drugs and immunomodulatory agents such as 5-fluorouracil (5-FU) [77,78,79,80], mitomycin C (MMC) [47,81,82,83,84,85,86,87], and interferon alfa (IFNα) [44,88,89,90,91,92] as monotherapy. These agents can treat the entire ocular surface, thus treating subclinical and microscopic disease [46]. The utility of topical chemotherapy ranges from chemo-reduction prior to surgery [47,89], primary treatment, to adjunctive treatment after surgery to reduce the risk of recurrence [6].

As primary therapy, topical chemotherapy has been shown to be comparable to surgery with similar efficacy and recurrence rates [44]. Karp [74] has suggested that topical chemotherapy may be preferred for large (>4 clock hours (Shields) [41] or ≥5 mm (AJCC) [75]), multifocal and recurrent lesions [88,89].

### 3.1. 5-Fluorouracil (5-FU)

5-FU is a pyrimidine analog that inhibits thymidine synthase, impairing DNA and RNA synthesis preferentially in cancer cells, thus preventing DNA replication and their proliferation. This mainly affects cells in the S stage of mitosis [42,93].

5-FU is typically applied as a topical ophthalmic formulation. Subconjunctival and perilesional injections have also been used in limited studies [94]. The most widely used protocol recommends 1% 5-FU drops four times daily for 1 week, followed by 3 weeks off as one cycle, for a total of 4 cycles [6]. Table 1 below summarises the effectiveness of different protocols where 5-FU was used as primary therapy.

Studies have shown 5-FU to be very effective as primary therapy for OSSN, with high-resolution rates of 82–100% and low recurrence rates of 6–14%. Its effectiveness has not been affected by patient age, gender, and ethnicity [80,96]. Compared to topical IFNα−2b, rates of tumour resolution, recurrence and time to response were similar with 5-FU [80,97,98].

5-FU is generally well tolerated with mostly mild side effects reported, including pain, redness, eyelid edema, tearing, keratopathy and superficial stromal melting [97]. Many of these adverse effects can be managed through regular use of preservative free artificial tears throughout the course of treatment and short-term use of topical steroids as needed. Punctal and canalicular stenosis have not been reported to arise from topical treatment, although they have resulted from systemic administration of 5-FU for other cancers [96,99].

Although topical 5-FU has fewer reported side effects compared to MMC [96,97], it is associated with more side effects compared to IFNα−2b as it also affects the proliferation of normal rapidly dividing epithelial cells and fibroblasts [89,100]. However, 5-FU generally costs less than IFNα−2b, despite the need for compounding [42]. While 5-FU can be stored at room temperature, refrigeration is recommended [80,101,102].

### 3.2. Mitomycin C (MMC)

MMC is an antimetabolite and alkylating agent that exerts its antineoplastic and antibiotic properties through inhibiting DNA synthesis and fibroblast migration as well as by inducing apoptosis. Compared to 5-FU, which mainly affects cells in the S stage of mitosis, MMC affects both proliferating and non-proliferating cells [89].

MMC is typically applied as topical eye drops. The most widely used protocol is 0.04% MMC drops four times daily for 1 week followed by 2–3 weeks off until the eye is quiet, for a total of 3 cycles, sometimes with punctual occlusion during administration [47,73,86,103,104]. However, various protocols have been reported with varying efficacies, as summarized in Table 2 below.

As primary therapy, studies have shown MMC to be very effective in treating OSSN, with high-resolution rates of 92–100% and low recurrence rates. Although the rates of tumour resolution were similar compared to IFNα−2b, MMC had a faster time to resolution of 1.5 months compared to 3.5 months [105].

However, use of MMC is limited by its toxicity. MMC has more frequent and severe ocular side effects compared to 5-FU and IFNα−2b [96,97,105]. These include ocular pain, redness, allergic conjunctivitis, tearing, epitheliopathy, conjunctival hyperaemia, punctate staining of the cornea, punctal stenosis and LSCD [81,82,87,103,105,106,107,108,109,110,111]. Topical preservative free tears and steroids are commonly used throughout the course of treatment to alleviate toxicity while punctal plugs may be used to prevent punctal stenosis [84,89,107,112]. Additionally, ancillary management with hyaluronic acid eye drops results in improvements in both subjective and objective ocular parameters, reducing MMC induced adverse effects [113]. Careful monitoring is needed for early identification of any signs of toxicity, as MMC administration should be halted if epitheliopathy occurs to minimise its toxic effects.

Other disadvantages of MMC include its need for compounding and refrigeration. Although it may be more costly than 5-FU, it is generally cheaper than IFNα−2b.

### 3.3. Interferon Alfa (INFα)

#### 3.3.1. Interferon Alfa-2b (INFα-2b)

IFNα−2b is a leukocyte derived low molecular weight glycoprotein that functions as an immunomodulatory cytokine. It exerts its anti-proliferative, anti-angiogenic and cytotoxic effects through activating pathways involved in the alteration of gene expression, apoptosis, inhibition of protein synthesis and inducing host anti-tumour immunosurveillance [114,115,116,117]. IFNα−2b also enhances the production of IL-2 and IFN-γ mRNA by the immune system and lowers the production of IL-10 which aids in the recognition and targeting of neoplastic cells [118]. Thus, immunocompetency is required for efficacious use of IFNα−2b [119,120]. The use of non-immunomodulating agents like 5-FU or MMC may be preferred in immunosuppressed patients.

IFNα−2b can be used as a topical eye drop or a subconjunctival perilesional injection. The most widely used protocol for topical drops is 1 million international units (MIU)/mL four times daily until resolution followed by at least one to three more months after resolution with a mean time to resolution of 4 months [6,42,121,122] while subconjunctival perilesional injections are typically given at a dose of 3MIU/0.5cc weekly [121] or 10 MIU/0.5cc monthly until resolution [123]. Table 3 below summarises the effectiveness of different protocols where IFNα−2b was used as primary therapy.

As primary therapy, both forms are highly effective in treating OSSN, with high rates of disease resolution of 81–100% and 87–100% and remarkably low recurrence rates of 0–5% and 0–7% for topical eye-drops and subconjunctival injections, respectively. A study comparing doses of topical IFNα−2b drops found that the 1 MIU/mL formulation is equally effective as the 3 MIU/mL dose with fewer side effects [122]. The effectiveness of both topical and subconjunctival IFNα−2b has not been affected by patient demographics [129].

Compared to 5-FU, the rates of tumour resolution, recurrence and time to response were similar [80,97,98]. When compared to MMC, rates of tumour resolution were also similar. However, time to response was longer for IFNα−2b, with median time to resolution of 3.5 months compared to 1.5 months for MMC. Regardless, IFNα−2b may be preferable to MMC as it is associated with significantly lower adverse effect rates of 12% compared to 88% [105].

Topical IFNα−2b drops are the best tolerated among the topical therapies for OSSN available [89,100,105]. Side effects of topical IFNα−2b are largely limited to mild irritation, conjunctival hyperaemia [52], reactive lymphoid hyperplasia [130] and follicular conjunctivitis [123] as it is an endogenous molecule as opposed to an external chemotherapeutic agent [125,131]. Most side effects resolve with discontinuation of treatment.

While also well tolerated, subconjunctival injections are associated with transient systemic flu-like symptoms such as fever, chills and malaise. However, these are easily alleviated with an oral antipyretic such as acetaminophen, and usually become less severe with repeated injections [121]. The advantages of subconjunctival injections include a faster time to resolution, lower cost, better compliance and greater accessibility as no compounding is necessary [121].

Similar to 5-FU, topical IFNα−2b drops require compounding. However, IFNα−2b requires refrigeration while 5-FU does not. IFNα−2b’s greatest limitation is its high cost and limited accessibility, although it may be less costly and more widely available in some countries, such as India. In addition, its frequent daily application until tumour resolution makes patient compliance a significant consideration when choosing to use topical IFNα−2b drops Thus, weekly or monthly subconjunctival IFNα−2b injections may be a better alternative to ensure compliance.

#### 3.3.2. Interferon Alfa-2a (INFα-2a)

IFNα-2b has been the most commonly used form of IFN in the treatment of OSSN. However, IFNα-2a is described in several case reports as a replacement for IFNα−2b, usually due to the limited accessibility and greater costs of IFNα-2b.

INFα-2a is cheaper and more readily available than IFNα-2b. The pegylated form of INFα-2a is more stable than IFNα-2b and has a longer half-life. When used, IFNα-2a has been administered both intralesionally and topically, both as monotherapy and as tumour reduction prior to surgery [50,51,132]. However, studies with larger sample sizes are required to substantiate the effectiveness of this product. The protocols used in several case reports are summarized in Table 4 below.

Side effects of IFNα−2a are reported to be mild and limited to ocular discomfort without long-term ocular complications [51,132]. However, evidence surrounding the effectiveness, recurrence rates and side effects of IFNα−2a in the treatment of OSSN remains limited.

## 4. Combined Medical and Surgical Management

In addition to primary therapy, topical chemotherapy may be used in combination with surgery in the form of chemo-reduction [47,89] or as adjunctive treatment to reduce the risk of recurrence [6].

Chemo-reduction may be useful in cases likely to respond poorly to topical monotherapy alone, although the evidence is limited as no studies have compared neoadjuvant chemotherapy to other modalities of treatment. Neoadjuvant chemotherapy refers to chemotherapy given to patients before their primary course of treatment. A case series by Shields et al. [47] found that thick tumours ≥ 4 mm may only show partial regression with MMC despite multiple chemotherapy cycles. Prolonged therapy increases the risk of MMC toxicity leading to serious vision and globe threatening complications [134]. Therefore, topical chemo-reduction followed by surgical resection was recommended, to allow for less extensive surgical resection and tissue reconstruction. The combination of topical neo-adjuvant chemotherapy followed by surgical excision may allow patients to benefit from both modalities by limiting the extent of surgical excision and its associated complications, while facilitating a tissue diagnosis and reducing the risk of tumour recurrence [83].

Meanwhile, there may be a role for systemic neoadjuvant chemotherapy (NAC) in the management of both intraorbital and extraorbital invasive OSSN. A retrospective case series found that systemic NAC was helpful in the management of surgically challenging cases of AJCC grade T3 and T4 OSSN and may be helpful in avoiding orbital exenteration [135].

Adjuvant topical chemotherapy reduces the postoperative risk of tumour recurrence, so termed *chemoprevention*. Factors associated with a higher risk of recurrence include positive pathological margins, tarsal involvement, higher grade lesions such as CIS and SCC, superior location of tumour, papillomatous morphology and recurrent OSSN [48].

Post-operative topical IFNα−2b therapy (1MIU/mL four times daily for two months) lowered the recurrence rate in patients with positive margins from 13% to 4%, a level comparable to rates reported in patients with negative margins [48]. Excisional biopsy followed by topical IFNα−2b therapy was reported to be the best modality of treatment to minimize tumour recurrence in a literature based analysis comparing surgery and topical IFNα−2b [136].

Similar to IFNα−2b, MMC has also been prescribed post-operatively in the management of positive margins, which decreased the incidence of recurrence from 66.7% to 5.9% [137]. Another study found a 0% recurrence rate in 27 eyes after a mean follow up period of 28 months with the use of postoperative MMC, which was sometimes combined with intraoperative cryotherapy [73]. MMC has also been used intraoperatively as an adjunct to surgical excision. Sarici et al. [59] found that excision of OSSN combined with cryotherapy and intraoperative MMC was effective with a low recurrence rate.

In a randomized, double-blind, placebo-controlled trial by Gichuhi et al. [138], 4 weeks of topical 1% 5-FU after surgical excision of OSSN substantially reduced the 1-year recurrence rate from 36% to 11%.

These studies suggest that surgical excision may not be sufficient to manage all cases of OSSN and highlight the efficacy of adjuvant topical chemotherapy in reducing rates of tumour recurrence, thus supporting their application in cases with high recurrence risk.

It is difficult to make a recommendation regarding the best treatment regime for chemo-reduction or chemoprevention given the lack of randomized controlled trials or comparative studies assessing the different drugs as well as their dosages and schedules.

A systematic review compared treatment results from prospective non-comparative series, found that adjuvant MMC and 5-FU produced similar effects in terms of disease control. Interestingly, MMC was reported to have a better toxicity profile [139].

The same study found limited evidence supporting the use of topical chemotherapy as adjuvant treatment rather than primary monotherapy and vice versa. The only comparative study was done with IFNα−2b, which found that surgical excison followed by adjuvant IFNα−2b resulted in a 100% response rate regardless of the tumour’s AJCC stage, compared to an 82% response rate with IFNα−2b as monotherapy. However, it gave a low level of evidence for this finding.

A retrospective study comparing the recurrence rate and the complications of surgical excision, surgical excision with adjuvant topical MMC, and surgical excision with adjuvant subconjunctival IFNα-2b found adjuvant IFN to have the lowest recurrence rates, with MMC causing ocular complications and IFN causing systemic symptoms [52]. Another retrospective study comparing adjuvant 5-FU and adjuvant MMC found one recurrence in the 5-FU group and no recurrences in the MMC group. Side effects occurred in 69% of the 5-FU patients and 41% of the MMC patients [140].

Further studies are warranted to elucidate the relative efficacy and safety of 5-FU, MMC and IFNα-2b as neoadjuvant or adjuvant topical chemotherapy. These considerations are likely to be similar to those discussed in Section 3.

## 5. Adjunctive Treatment

A range of alternative treatments have been used sporadically in the management of OSSN. These include photodynamic therapy [141], anti-VEGF agents [55,142,143,144,145], cidofovir [56,57], aloe vera [146], retinoic acid [54,147], proton beam therapy [148,149,150,151,152], plaque brachytherapy [153,154,155,156,157,158], electron beam radiotherapy [159], phototherapeutic keratectomy with excimer laser [160], urea [161], dinitrochlorobenzene [162] and laser photocoagulation [163]. Although these are not primary treatment modalities, they have been investigated for adjunctive management of OSSN.

### 5.1. Photodynamic Therapy (PDT)

PDT utilizes verteporfin, a photosensitive dye with a high affinity for abnormal blood vessels, that causes apoptosis and immunological destruction of neighbouring structures when exposed to light of specific wavelengths. Barbazetto et al. [141] treated three patients with SCC of the conjunctiva with one to three treatments of verteporfin (6 mg/m^2^ body surface area, intravenously) with a light dose of 50 J/cm^2^ 1 min after the injection. In this study, PDT was found to be effective in causing localized tumour regression with minimal side effects. This limited evidence suggests that PDT may be a useful adjunct in OSSNs localized to the conjunctiva.

### 5.2. Anti-Vascular Endothelial Growth Factor (Anti-VEGF)

Anti-VEGF agents are monoclonal antibodies that target VEGF to inhibit angiogenesis. Although VEGF has been detected in conjunctival SCC [164], topical bevacizumab and ranibizumab have had variable success in the treatment of OSSN [55,142,143,144,145]. As studies are limited to a small number of case series and case reports with small sample sizes, there is little evidence to support the efficacy or use of these agents. In one case series of 6 patients, topical bevacizumab was shown to be effective as neoadjuvant therapy combined with surgical excision for the treatment of OSSN, with surgical excision unnecessary in responsive patients [55]. Another case series of 10 patients found perilesional or subconjunctival injections of bevacizumab to be potentially curative in lesions limited to the conjunctiva. However, there was no significant effect in cases with corneal extension of OSSN [142]. There is limited evidence that subconjunctival ranibizumab may be effective in treating extensive SCC refractory to treatment with all other standard agents [144]. However, other studies have not found anti-VEGF agents to be effective in the treatment of OSSN [143], thus their application in clinical practice is questionable. Anti-VEGF agents have been reported to be well tolerated, with side effects limited to mild irritation and epiphora [55,142,143,144,145].

### 5.3. Cidofovir

Cidofovir is an antiviral that acts against double-stranded DNA viruses such as HPV. Topical cidofovir has shown to be effective in a small number of patients for the treatment of refractory OSSN in the short to medium term, especially in the presence of a high-risk HPV serotype (HPV-16), and its efficacy raises the possibility of a viral etiology of OSSN in patients resistant to IFNα-2b [56,57]. However, the antitumour effects of cidofovir independent of HPV presence is unknown and comparative clinical trial data lacking.

### 5.4. Aloe Vera

Aloe Vera in its topical form may be useful in the treatment of OSSN. This is illustrated in an isolated case report where OSSN completely resolved after 3 months of topical aloe vera treatment [146]. However, this case was not biopsy proven as the patient refused biopsy. Further research is required to determine if there is a role for aloe vera in OSSN management.

### 5.5. Retinoic Acid

Retinoic acid is a metabolite of vitamin A that mediates its functions and plays important roles in cell growth and differentiation. In a case series of 4 patients, topical retinoic acid alone was found to be effective in treating severe squamous metaplasia in cicatrizing diseases of the conjunctiva [147]. Another study found that treatment with topical retinoic acid and IFNα−2b was more effective in treating lesions with minimal self-limited side effects, faster and greater resolution and a longer tumour-free period compared to IFNα−2b alone [54]. This suggests that the combination of topical retinoic acid and IFNα−2b may be a superior alternative to topical IFNα−2b alone in the treatment of CIN. Thus, retinoic acid may be useful as complementary therapy, especially in patients who are not keen on either surgery or injections.

### 5.6. Radiotherapy

Various forms of radiotherapy including proton beam therapy [148,149,150,151,152], plaque brachytherapy with strontium-90 (Sr-90), ruthenium-106 (Ru-106) and iodine-125 (125I) [153,154,155,156,157,158] and electron beam radiotherapy [159] have been used in the management of OSSN, especially in cases of invasive disease. Plaque radiotherapy may be useful as adjuvant treatment in scleral-invasive SCC [153,154,155,156,157,158], while proton beam therapy or electron beam radiotherapy may be useful in the treatment of recalcitrant SCC of the conjunctiva [148,149,150,151,152,159]. Recently, histopathology-guided use of Ru-106 surface plaque brachytherapy was effectively used as adjuvant therapy in the management of OSSN with corneal stromal and scleral invasion [157].

## 6. Approach to the Management of OSSN

No guidelines exist to advise an approach to the management of OSSN. Based on available evidence an algorithm was developed (Figure 1, Figure 2, Figure 3 and Figure 4) to summarize the modalities available for treatment and proposes an approach to management. This algorithm may provide ophthalmologists a guide to assist selecting the most appropriate treatment regime based on both patient and disease factors to minimize treatment related morbidity and improve OSSN control.

Diagnosis of OSSN is suggested by clinical examination and investigations including histopathological evaluation following an incisional or excisional biopsy, which is the gold standard to confirm the diagnosis of OSSN [33]. Less invasive modalities such as impression [35] or exfoliative cytology [36,37] are limited to the identification of superficial dysplastic lesions. In vivo confocal microscopy (IVCM) assessment for dysplasia or neoplasia [38,39] and high-resolution or ultra-high-resolution anterior segment optical coherence tomography (HR-OCT) can identify thickened (>120 microns) hyperreflective epithelium with an abrupt transition zone [40]. If the diagnosis is uncertain, HR-OCT and impression cytology can be performed to add to the evidence for the diagnosis before an excisional biopsy is performed.

Although surgical management is currently the gold standard for the management of OSSN and potentially shortens the overall treatment period compared to medical treatment, the complications highlighted in previous sections limit its wider application. Such that there has been a shift towards medical management [76] including topical chemotherapy drugs and immunomodulatory agents including 5-fluorouracil (5-FU) [77,78,79,80], mitomycin C (MMC) [47,81,82,83,84,85,86,87], and interferon alfa-2b (IFNα−2b) [44,88,89,90,91,92] as monotherapy alone due. These topical agents can treat the entire ocular surface enabling management of subclinical and microscopic disease [46].

As proposed by Karp [74], surgery may be preferred for small (<4 clock hours (Shields) [41] or ≤5 mm (AJCC) [75]), unifocal lesions, lesions with an uncertain diagnosis and first presentations of OSSN while topical chemotherapy may be preferred for large (>4 clock hours (Shields) [41] or ≥5 mm (AJCC) [75]), multifocal and recurrent lesions [88,89]. Considerations of patient factors such as affordability, compliance and patient co-morbidities and preference also influence the choice between surgical and medical management. Surgical management is preferred in patients with contraindications to topical ophthalmic chemotherapy. For example, patients receiving systemic treatment for systemic comorbidities may experience ocular side effects that limit the use of topical chemotherapy. In addition, the presence of significant medical comorbidities necessitating polypharmacy may reduce patient compliance to topical medical therapy, especially since most protocols involve a high frequency of dosing (four times a day) over an extended duration of several months. Such sustained compliance is difficult to achieve in the elderly, who make up the majority of OSSN patients.

OSSNs undergoing surgical management are excised primarily via Shields’ no touch technique (4 mm margins) followed by intraoperative cryotherapy with the double freeze slow thaw technique [41]. Kerato-epitheliectomy can be performed in cases of corneal involvement [41] while a partial lamellar sclerectomy can be performed for scleral invasion [61]. In rarer cases of intraocular invasion, enucleation [62] and plaque radiotherapy [153,154,155,156,157] can be considered while orbital invasion requires exenteration [63].

It is recommended that excised conjunctival specimens be sent unrolled and mounted onto filter paper before placing in formalin fixation. Orientation can be marked on the filter paper to allow accurate margin assessment. Larger specimens may be further oriented with sutures in case of dislodgment from filter paper. If histopathological evaluation shows positive margins, adjunctive topical chemotherapy with IFNα−2b, 5-FU or MMC should be performed to reduce recurrence rates. Adjunctive topical chemotherapy can also be considered with negative margins given the presence of factors associated with a higher risk of recurrence including tumours with tarsal involvement, higher grade lesions such as CIS and SCC, superior location of tumour, papillomatous morphology and recurrent OSSN [48].

5-FU, IFNα−2b and MMC are the three main topical monotherapies for OSSNs undergoing medical management (Figure 3). In an isolated case report, topical pegylated interferon-alpha-2a (peg-IFNα-2a) was effectively used as an immunotherapeutic agent for the management of OSSN with no ocular surface toxicity. Thus, it may be a useful adjunct to the main topical monotherapies, although more studies are required to confirm its efficacy [51].

In immunosuppressed individuals including those on steroids, immunosuppressants, or with conditions such as leukemia and multiple myeloma, the use of non-immunomodulating agents like 5-FU or MMC may be preferred [119,120]. In most other cases, 5-FU and IFNα−2b would be preferred to avoid the toxic side effects of MMC, with MMC reserved as a last resort for recalcitrant cases. In these cases, patients should be carefully monitored for early identification of any signs of toxicity, as MMC administration should be halted if epitheliopathy occurs to prevent further toxic effects. The choice between 5-FU and IFNα−2b is dependent on logistical considerations and patient factors. 5-FU is the only drug of the three that does not require refrigeration [80,101,102] and is generally the most cost-effective option despite compounding. IFNα−2b has the best side effect profile, however its use requires greater patient compliance and is limited by its high cost and concerns of accessibility. Between topical IFNα−2b drops and subconjunctival injections, the latter is less costly, more accessible, allows for a faster time to resolution and is less demanding in terms of patient compliance, with the trade-off of more systemic flu-like symptoms such as fever, chills and malaise after administration [121]. Patient preference is likely the deciding factor in choosing between the two forms of IFNα−2b.

Sequential use of or a combination of topical chemotherapeutic drugs may be considered to improve response to therapy and to reduce rates of recurrence if factors associated with poor response to topical monotherapy alone are present. These include multifocality [96], thick tumours (≥4 mm (MMC), >1.5 mm (5-FU)) [47,96], nasal location of tumour [80], circumference of ocular surface involved > 6 clock hours [129] and larger tumour diameter ≥ 21 mm [129]. Combination chemotherapy may also be preferred in time sensitive conditions to expedite response to therapy [129] and in smokers to reduce recurrence rates [165].

Alternatively, a combination of surgical and medical methods such as neoadjuvant debulking followed by surgery or adjuvant topical chemotherapy may be considered.

A range of alternative treatments have been used sporadically in the management of OSSN. These include photodynamic therapy [141], anti-VEGF agents [55,142,143,144,145], cidofovir [56,57], aloe vera [146], retinoic acid [54,147], proton beam therapy [148,149,150,151,152], plaque brachytherapy [153,154,155,156,157,158], electron beam radiotherapy [159], phototherapeutic keratectomy with the excimer laser [160], urea [161], dinitrochlorobenzene [162] and laser photocoagulation [163]. Although these are not primary treatment modalities, limited evidence suggests that they may be useful in the adjunctive management of OSSN. PDT and anti-VEGF agents may be useful adjuncts in OSSNs localized to the conjunctiva. Topical retinoic acid may be useful as complementary therapy with IFN-α-2b. Adjuvant radiotherapy may play a role in the management of invasive disease with plaque radiotherapy being used for scleral-invasive SCC [153,154,155,156] and proton beam therapy or electron beam radiotherapy being used for recalcitrant SCC of the conjunctiva [148,149,150,151,152,159]. Further studies are needed to establish the efficacy and safety of these agents.

Following medical or surgical treatment, high-resolution (5–10 µm) or ultra-high-resolution (3–5 µm) anterior segment optical coherence tomography (HR-OCT) should be performed to confirm resolution of OSSN. HR-OCT has been found to be highly sensitive and specific in the diagnosis and treatment of OSSN [40], and detected subclinical OSSN in 17% of cases with apparent clinical resolution in one study [166]. HR-OCT may be used to prevent premature termination of treatment [167] and to monitor the progression of treatment to reduce the risk of overtreatment with topical chemotherapy, potentially resulting in cost savings and improved safety for patients. If HR-OCT is unavailable or if subclinical disease is present, additional cycles of topical chemotherapy should be performed until resolution. After resolution, HR-OCT may be used for surveillance to monitor for subclinical recurrences during regular follow ups.

## 7. Future Developments

The growing popularity of primary topical monotherapy with chemotherapeutic drugs and immunomodulatory agents represents a shift towards medical management of OSSN, which may replace surgery as the standard of care in the future [46]. However, the lack of randomized controlled trials or comparative studies assessing the different drugs as well as their dosages and schedules makes it difficult to recommend best practice guidelines. Further studies, especially prospective randomized trials, are warranted to evaluate the use of these drugs, their dosages, schedules, and concentrations in the treatment of OSSN and compare the outcomes to surgical management. More work can also be done to elucidate the role of combination topical chemotherapy in OSSN management as well as the relative efficacy and safety of 5-FU, MMC and IFNα-2b as neoadjuvant or adjuvant topical chemotherapy.

HR-OCT may assist both the medical and surgical management of OSSN. Studies evaluating the efficacy of HR-OCT in diagnosing OSSN, monitoring treatment progression and resolution, as well as in diagnosing recurrences may support its wider application in clinical practice. The integration of HR-OCT with operating microscopes may improve delineation of tumour margins to aid complete tumour resections, thus reducing the risk of recurrence.

## 8. Conclusions

The rise of primary topical monotherapy with chemotherapeutic drugs and immunomodulatory agents has improved the management of OSSN, especially in recurrent or extensive disease and in minimizing morbidity associated with surgery. Complemented by HR-OCT, medical management holds promise for the treatment of complex ocular surface disease. The best modality of treatment is patient-tailored carefully selected on the basis of disease factors, co-morbidities, logistical factors including the need for refrigeration, compounding, accessibility, cost and the input of a well-informed, compliant patient. The algorithm in this paper summarizes the modalities available for treatment and proposes an approach to management in an attempt to guide ophthalmologists in selecting the most appropriate treatment regime to minimize treatment related morbidity and improve OSSN control. Further work can be done to validate this algorithm and to develop formal guidelines to direct the management of OSSN.

## Figures and Tables

**Figure 1 ijms-24-00713-f001:**
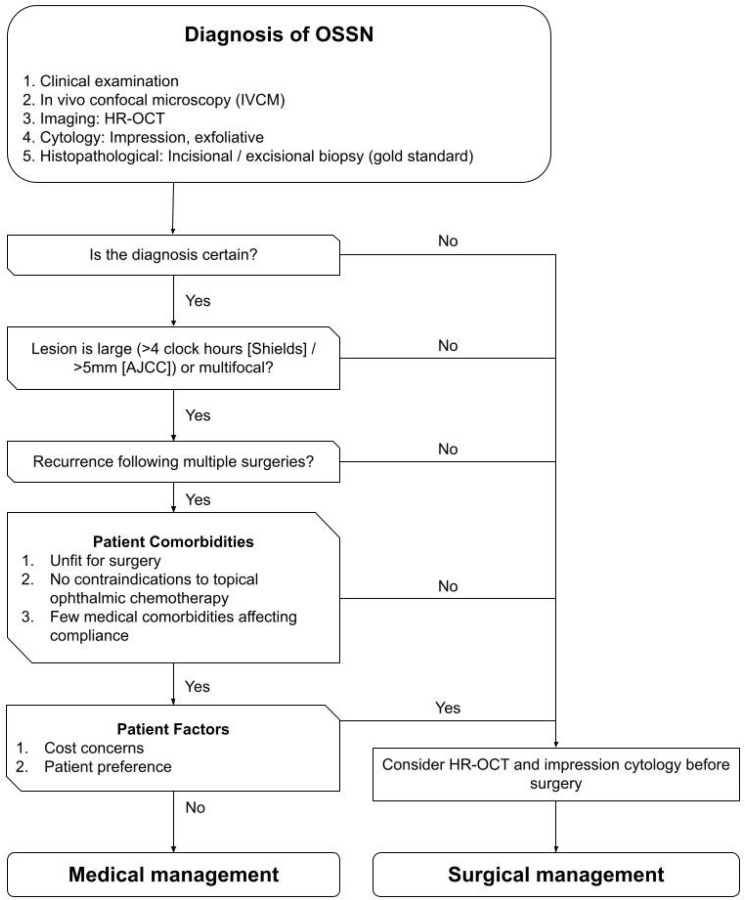
Flowchart guiding the choice between medical and surgical management of ocular surface squamous neoplasia (OSSN).

**Figure 2 ijms-24-00713-f002:**
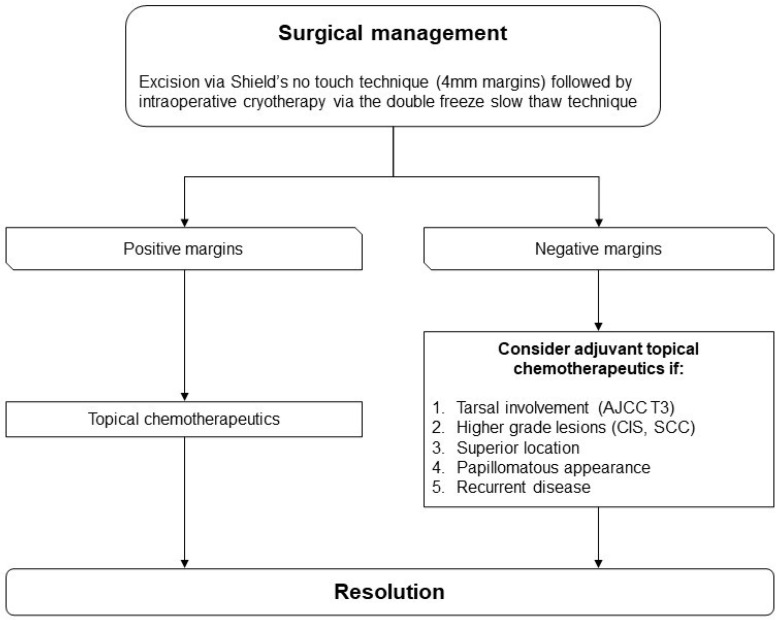
Flowchart guiding the choice of adjuvant topical chemotherapy following surgical management of OSSN.

**Figure 3 ijms-24-00713-f003:**
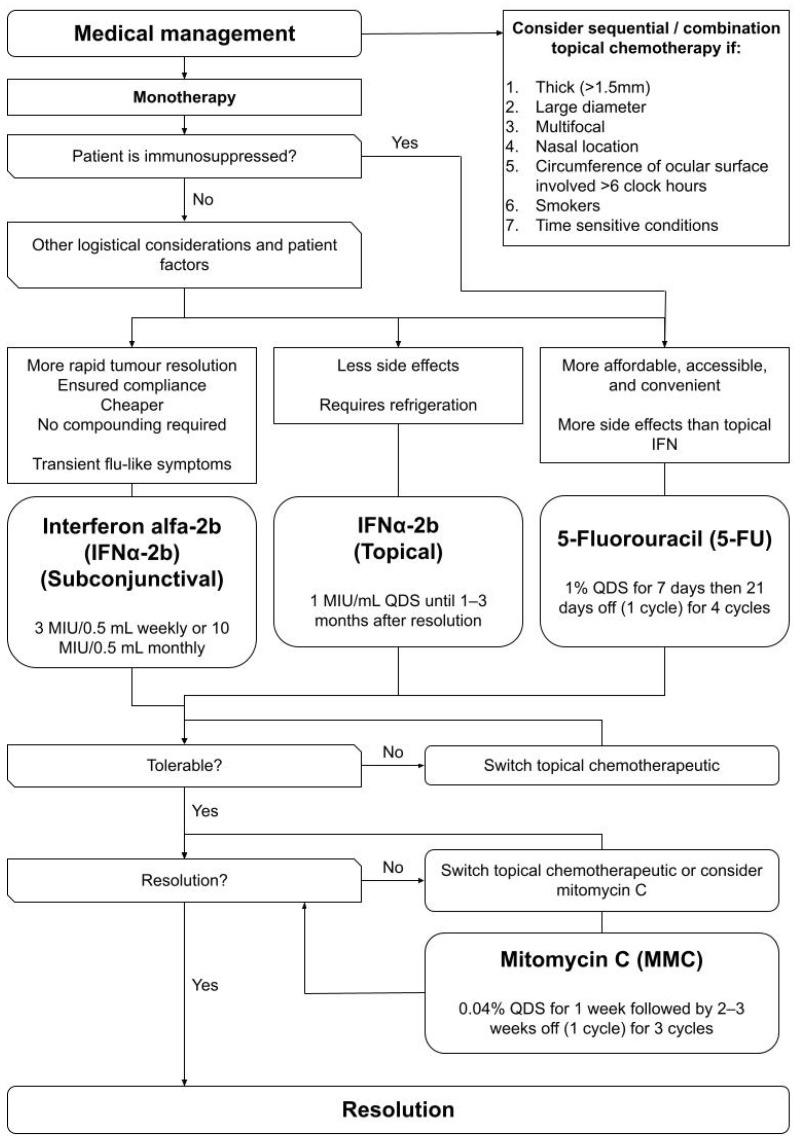
Flowchart guiding the choice of topical chemotherapy as primary medical therapy for OSSN, and the need for sequential or combination topical chemotherapy.

**Figure 4 ijms-24-00713-f004:**
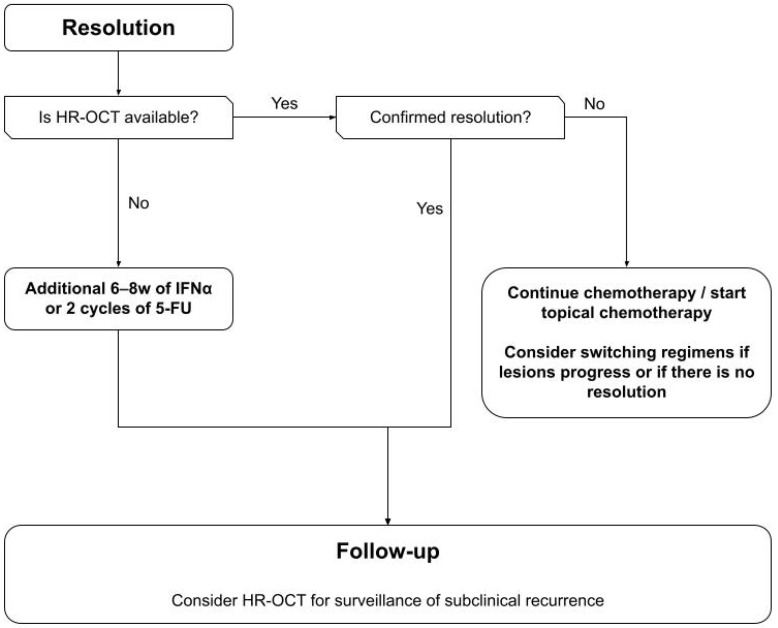
Flowchart guiding the monitoring of OSSN resolution and surveillance of disease recurrence.

**Table 1 ijms-24-00713-t001:** Summary of study protocols using 5-Fluorouracil (5-FU) as sole treatment of ocular surface squamous neoplasia (OSSN).

Study	Protocol	Outcomes
Topical 1% 5-fluorouracil in ocular surface squamous neoplasia: a long-term safety study [95] **Study location:** Italy **Number of patients:** 22	**Dose:** 1% 5-FU topical QDS for 4 weeks, adjunctive courses administered after 3 months of chemotherapy-free interval Eye ointment applied to inferior eyelid to minimise skin contact, and inferior lacrimal punctum briefly occluded **Endpoint:** Until clinical and cytological tumour regression	**Response:** 22 (100%) **Time to response:** Not specified **Recurrence:** 3 (14%) **Side effects:** not reported
Topical 5-Fluorouracil 1% as Primary Treatment for Ocular Surface Squamous Neoplasia [80] **Study location:** United States **Number of patients:** 44	**Dose:** 1% topical 5-FU QDS for 1 week followed by a drug holiday of 3 weeks No punctal plugs **Endpoint:** Until clinical resolution or failure to respond within 4 or more cycles	**Response:** 36 (82%) **Time to response:** 3.8 cycles (median: 4, range: 2–9) **Recurrence:** 4 (6% at 1 y, 15% at 2 y) **Side effects:** pain (17, 39%), tearing (10, 23%), photophobia (6, 14%), itching (4, 9%), swelling (2, 5%), infection (1, 2%), no long-term complications
Topical 1% 5-fluoruracil as a sole treatment of corneoconjunctival ocular surface squamous neoplasia: long-term study [96] **Study location:** Italy **Number of patients:** 41	**Dose:** 1% 5-FU topical QDS for 4 weeks, adjunctive courses administered after 1 month of chemotherapy-free interval Eye ointment applied to the inferior eyelid skin to minimise skin contact, no punctal occlusion **Endpoint:** Until clinical resolution or clinical evidence of lack of further tumour response	**Response:** 34 (83%) **Time to response:** 11 ± 9 weeks (range: 3–22 weeks) **Recurrence:** 4 (9%) 2 were treated with topical MMC and 2 were treated with topical MMC + surgery, all responded completely. **Side effects:** photophobia (20, 51%), conjunctival hyperemia (19, 48%), irritation (17, 43%), pain (14, 36%), superficial punctate keratitis (11, 28%), lid erythema (4, 8%)
Comparison of Topical 5-Fluorouracil and Interferon Alfa-2b as Primary Treatment Modalities for Ocular Surface Squamous Neoplasia [97] **Study location:** United States **Number of patients:** 54	**Dose:** 1% topical 5-FU QDS for 1 week followed by a drug holiday of 3 weeks **Endpoint:** Until clinical resolution or failure to respond within 2 cycles	**Response:** 52 (96%) **Time to response:** 6.6 ± 4.5 **Recurrence:** 6 (12%), mean of 7.7 ± 9.1 months **Side effects:** pain (12, 22%), tearing (12, 22%), redness (11, 20%), eyelid edema (5, 93%), keratopathy (4, 7%), no long-term complications

**Table 2 ijms-24-00713-t002:** Summary of study protocols using Mitomycin C (MMC) as sole treatment of OSSN.

Study	Protocol	Outcomes
Randomized controlled trial of topical mitomycin C for ocular surface squamous neoplasia: early resolution [87] **Study location:** Australia **Number of patients:** 26 with MMC vs. 22 with placebo	**Dose:** MMC 0.4 mg/mL QDS for 3 weeks Advised punctal plugs or manual occlusion **Endpoint:** Until resolution by slit lamp examination or failure to regress by 6 weeks	**Response:** 24 (92%) **Time to response:** 121 days (range: 73–169) **Recurrence:** Not specified **Side effects:** redness, irritation
Long-Term Results of Topical Mitomycin C 0.02% for Primary and Recurrent Conjunctival-Corneal Intraepithelial Neoplasia [103] **Study location:** Brazil **Number of patients:** 18	**Dose:** 0.02% topical MMC QDS for 14 days, followed by 14 more days if no resolution Manual punctal occlusion for 3 min **Endpoint:** For 28 consecutive days	**Response:** 18 (100%) **Time to response:** 28 days **Recurrence:** 0 (0%) **Side effects:** conjunctival hyperemia, tearing, corneal epithelial erosions (2, 11%)
Retrospective Comparative Study of Topical Interferon α2b Versus Mitomycin C for Primary Ocular Surface Squamous Neoplasia [105] **Study location:** India **Number of patients:** 25	**Dose:** MMC 0.4 mg/mL QDS in 1 week on and 1 week off cycles Advised punctal occlusion for 5 min during treatment course **Endpoint:** Until clinical resolution or failure to regress after 2 cycles	**Response:** 23 (92%) **Time to response:** 1.5 ± 0.54 (median 1.5) **Recurrence:** 0 (0%) **Side effects:** Conjunctival hyperemia (11, 44%), hyperemia with burning sensation (9, 36%), corneal epitheliopathy (3, 12%), photophobia with blepharospasm (2, 8%), punctal stenosis (1, 4%)

**Table 3 ijms-24-00713-t003:** Summary of study protocols using Interferon alfa-2b (IFNα−2b) as sole treatment of OSSN.

Study	Protocol	Outcomes
**Topical**		
Regression of presumed primary conjunctival and corneal intraepithelial neoplasia with topical interferon alpha-2b [124] **Study location:** United States **Number of patients:** 7	**Dose:** 1 MIU/mL IFNα−2b 4–6 times daily **Endpoint:** Until 1 month beyond complete clinical resolution via slit-lamp biomicroscopy	**Response:** 7 (100%) **Time to response:** median: 54 days, mean: 77 days, range: 28–188 days **Recurrence:** 0 (0%) **Side effects:** conjunctival hyperemia and follicular conjunctivitis (4, 57%)
Topical Interferon α-2b as a Single Therapy for Primary Ocular Surface Squamous Neoplasia [125] **Study location:** India **Number of patients:** 24	**Dose:** 1 MIU/mL topical IFNα−2b QDS **Endpoint:** Until clinical resolution or failure to regress within 3 months	**Response:** 22 (92%) **Time to response:** median: 3.25, range: 2–4 **Recurrence:** 0 (0%) **Side effects:** intratumoural bleeding (2, 8%), conjunctival congestion (1, 4%), foreign body sensation (1, 4%)
Retrospective Comparative Study of Topical Interferon α2b Versus Mitomycin C for Primary Ocular Surface Squamous Neoplasia [105] **Study location:** India **Number of patients:** 26	**Dose:** 1 MIU/mL topical IFN IFNα−2b QDS **Endpoint:** Until clinical resolution or failure to regress after 2 months	**Response:** 23 (89%) **Time to response:** 3.1 ± 0.73 (median 3.5) **Recurrence:** 1 (4%) at 18 m **Side effects:** conjunctival hyperemia (2, 8%), hyperemia with burning sensation (1, 4%)
Comparison of Topical 5-Fluorouracil and Interferon Alfa-2b as Primary Treatment Modalities for Ocular Surface Squamous Neoplasia [97] **Study location:** United States **Number of patients:** 48	**Dose:** 1 MIU/mL topical IFNα−2b QDS **Endpoint:** Not specified	**Response:** 39 (81%) **Time to response:** 5.5 ± 2.9 **Recurrence:** 2 (5%), mean of 9.9 ± 11.4 months **Side effects:** pain (9, 20%), redness (6, 13%), blurred vision (6, 13%), tearing (2, 4%), no long-term complications
Primary treatment of ocular surface squamous neoplasia with topical interferon alpha-2b: Comparative analysis of outcomes based on original tumor configuration [126] **Study location:** United States **Number of patients:** 61	**Dose:** 1 MIU/mL topical IFNα−2b QDS **Endpoint:** Until biomicroscopic evidence of tumour resolution or until the time a secondary treatment was deemed necessary due to poor response	**Response:** 59 (95%) complete response; 2 (3%) had partial response; additional treatment required for complete response in 7 (11%) **Time to response:** 5.8 (median: 5, range: 1–17.8) **Recurrence:** 2 (3%) **Side effects:** follicular reaction (4, 6%), corneal epithelial defect (2, 3%), irritation (1, 2%)
Recombinant Interferon Alpha-2b as Primary Treatment for Ocular Surface Squamous Neoplasia [127] **Study location:** Iran **Number of patients:** 92 * Data was combined with perilesional IFNα−2b used in some patients as second line	**Dose:** 3 MIU/mL topical IFNα−2b QDS **Endpoint:** Until 1 month beyond complete tumour resolution then tapered to BD for 2 months, or until failure to regress tumour in 2 subsequent monthly visits	**Response:** 89 (97%); 8 required perilesional IFNα−2b **Time to response:** 4.64 ± 1.92 months (median: 5, range: 1–10) **Recurrence:** Not specified **Side effects:** conjunctival hyperemia (4, 4%), follicular reaction (2, 2%), punctate epithelial erosions (1, 1%), chemosis (1, 1%)
**Subconjunctival**		
Management of Ocular Surface Squamous Neoplasia with Topical and Intralesional Interferon Alpha 2B in Mexicans [128] **Study location:** Mexico **Number of patients:** 6	**Dose:** 3 MIU/0.5 mL intralesional IFNα−2b once weekly **Endpoint:** Until resolution	**Time to response:** 6.5 months, range: 4–11 months **Recurrence:** 0 (0%) **Side effects:** None reported
Subconjunctival/Perilesional Recombinant Interferon α2b for Ocular Surface Squamous Neoplasia: A 10-Year Review [121] **Study location:** United States **Number of patients:** 15 * Some eyes were recurrences of OSSN	**Dose:** 3 MIU/0.5 mL perilesional subconjunctival IFNα−2b weekly (10 eyes had concomitant topical IFN, but the study found no difference in resolution) **Endpoint:** Until resolution	**Response:** 13 (87%) **Time to response:** median: 1.4 months, range 0.6–5.7 **Recurrence:** 1 (7%) **Side effects:** stinging and irritation (4, 27%), fever and malaise (5, 33%)
Perilesional and topical interferon alfa-2b for conjunctival and corneal neoplasia [92] **Study location:** United States **Number of patients:** 6 * All eyes had concomitant topical IFNα−2b	**Dose:** 3 MIU/0.5 mL perilesional subconjunctival IFNα−2b once followed by 1MIU/mL topical IFNα−2b QDS **Endpoint:** Until 1 month after clinical resolution	**Response:** 6 (100%) **Time to response:** within 6 weeks **Recurrence:** 0 (0%) **Side effects:** fever and myalgia (2, 33%)

**Table 4 ijms-24-00713-t004:** Case reports describing the use of Interferon alfa-2a (IFNα−2a) in the treatment of OSSN.

Study	Protocol	Outcomes
Interferon alpha-2a as alternative treatment for conjunctival squamous cell carcinoma [132] **Study location:** Peru **Number of patients:** 1	**Dose:** 1 MIU/mL topical IFNα−2a, one drop every 6 h **Endpoint:** For 4 months	Complete tumour resolution at the end of treatment No recurrence at 24 months
Immunoreduction of ocular surface tumours with intralesional interferon alpha-2a [50] **Study location:** United Kingdom **Number of patients:** 3 * IFNα−2a was used as tumour reduction prior to surgery	**Dose:** 3 MIU intralesional IFNα−2a 28d prior to surgery	Reduction in size and inflammation clinically and histological regression at surgery No side effects reported
Pegylated interferon-alpha-2a for the treatment of ocular surface squamous neoplasia [133] **Study location:** South Korea **Number of patients:** 8	**Dose:** 180 mcg / 0.5 mL intralesional pegylated IFNα−2a injection initially and 4 weeks later Together with 36 mcg/mL topical pegylated IFNα−2a QDS until 8 weeks following clinical resolution	Complete resolution after 12 weeks of initiation of treatment No recurrence at 6 months No side effects reported

## Data Availability

Data sharing is not applicable to this article.

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
