# Peer review of "The Management of Ocular Surface Squamous Neoplasia (OSSN)"

_ijms, 2022, doi:10.3390/ijms24010713_

Round 1

Reviewer 1 Report

Thank you for the review.

The tables that attempt to summarize the past literature regarding medical treatment of OSSN are difficult to follow.  Consider putting them in a different format.

Much of the information in part 6. Management of OSSN is repeated from the previous sections.  This part could be re-worked to provide new content or the text largely removed and more explanation of the flow charts could be included.

More detail regarding diagnosis of OSSN with in vivo confocal microscopy and HR-OCT could be included earlier in the paper. 

Is there new material being presented in this review as compared to other reviews by cited authors?  It was unclear what the reader should gain from this review although the extensive citation list is helpful.

Reviewer 2 Report

I have reviewed the paper entitled "The management of ocular surface squamous neoplasia" which is a review. It is a good work, my specific comments as:

- delete sentence "located in the  interpalpebral fissure", line 70.

- line 222,you could add the sentence "hyaluronic acid allows to improve both subjective and objective ocular parameters, , reducing MMC induced adverse effects" (Sammarco et al DOI: 10.1007/s00432-022-04241-5 )

- line 423, you should make reference to the use of strontium-90 and iodine-125 brachytherapy, as adjuvant therapy.

The work is very well done and takes a comprehensive approach. To make it more comprehensive and updated, I suggested a few minor changes to the authors.

Round 2

Reviewer 1 Report

The adjustments to the tables are helpful in improving the reader's ability to follow the treatment and endpoints.